# Is First Trimester Maternal 25-Hydroxyvitamin D Level Related to Adverse Maternal and Neonatal Pregnancy Outcomes? A Prospective Cohort Study among Malaysian Women

**DOI:** 10.3390/ijerph17093291

**Published:** 2020-05-08

**Authors:** Nor Haslinda Abd Aziz, Noor Azyani Yazid, Rahana Abd Rahman, Norhashima Abd Rashid, Sok Kuan Wong, Nur Vaizura Mohamad, Pei Shan Lim, Kok-Yong Chin

**Affiliations:** 1Department of Obstetrics and Gynaecology, Faculty of Medicine, Universiti Kebangsaan Malaysia Medical Centre, Cheras 56000, Malaysia; norhaslinda.abdaziz@ppukm.ukm.edu.my (N.H.A.A.); yanieyazid83@gmail.com (N.A.Y.); drrahana@ppukm.ukm.edu.my (R.A.R.); shimaabdrashid5@gmail.com (N.A.R.); pslim@ppukm.ukm.edu.my (P.S.L.); 2Department of Pharmacology, Faculty of Medicine, Universiti Kebangsaan Malaysia Medical Centre, Cheras 56000, Malaysia; jocylnwsk@gmail.com (S.K.W.); vaizuramohd@gmail.com (N.V.M.)

**Keywords:** vitamin D, first trimester, pregnancy complications, neonates

## Abstract

Information on the role of 25-hydroxyvitamin D (25(OH)D) in preventing adverse pregnancy/neonatal outcomes is limited in Malaysia. This study aims to determine the relationship between the level of maternal 25(OH)D in the first trimester of pregnant women and their pregnancy/neonatal outcomes. A total of 60 pregnant women in the first trimester were recruited and followed until the end of their pregnancy. The occurrence of any antenatal, delivery, and neonatal complications was recorded. Their blood was collected in the first trimester for total serum 25(OH)D determination using enzyme-linked immunosorbent assay. Overall, 10% of the women had vitamin D deficiency, while 57% had vitamin D insufficiency in their first trimester. No statistically significant difference in 25(OH)D level/status was observed between women with or without antenatal and delivery complications (*p* > 0.05). No difference in maternal serum 25(OH)D level and vitamin D status was observed between neonates with or without complications (*p* > 0.05). In conclusion, there is a high prevalence of vitamin D insufficiency among Malaysian pregnant women, but it is not associated with adverse maternal and neonatal outcomes. More comprehensive studies should be planned to verify this relationship.

## 1. Introduction

Vitamin D plays a key role in regulating bone mineralization and homeostasis of calcium and phosphorus [1]. Cutaneous synthesis stimulated by sunlight, dietary and supplementary consumptions are the primary sources of vitamin D [2]. It is then transported to the liver and is hydroxylated to 25-hydroxyvitamin D (25(OH)D) [3], which is the main circulating form of vitamin D in the human body [4]. Then, 25(OH)D is hydroxylated into 1-α-25-dihydroxyvitamin D (1,25(OH)D) in the kidney. Despite being the most active biological form of vitamin D, it has a short half-life, making it a poor indicator of vitamin D status [5]. Additionally, it is regulated by parathyroid hormone, calcium, and phosphate. Hence, its levels decrease only in severe vitamin D deficiency [6].

The Institute of Medicine (IOM) defines vitamin D deficiency as a serum level of 25(OH)D <30 nmol/L, and insufficiency as a level of 30–50 nmol/L [7]. Vitamin D deficiency has been identified as a public health problem among pregnant women. The prevalence of vitamin D deficiency and insufficiency during pregnancy ranges from 27% to 91% in the United States, 39% to 65% in Canada, 45% to 100% in Asia, 19% to 96% in Europe, and 25% to 87% in Australia and New Zealand [8]. Tropical countries do not escape from the problem of vitamin D deficiency [9]. Despite receiving perennial sunlight, vitamin D deficiency is prevalent among Malaysian pregnant women. A recent study conducted at a tertiary hospital in Kuala Lumpur found that 71.7% of third trimester pregnant women had vitamin D deficiency and 21% had vitamin D insufficiency [10]. Apart from its classical role in bone metabolism, vitamin D is involved in many other aspects of maternal and neonatal health. Vitamin D receptors, which mediate the biological activities of 1,25(OH)D, are found in many tissues, including the placenta [11]. Vitamin D deficiency is linked to a higher incidence of preeclampsia, gestational diabetes, bacterial vaginosis in mothers, preterm birth, and small for gestational age in newborns [12]. Vitamin D affects maternal and neonatal health by influencing gene expression related to immunity and inflammation in mothers, and DNA methylation and epigenetic regulation in children [13]. Some researchers believe that women need to enter pregnancy with sufficient 25(OH)D levels to prevent complications like preeclampsia, and later stage supplementation is futile in preventing pregnancy complications [13]. However, data on the role of vitamin D in preventing maternal and neonatal complications are very limited in Malaysia.

This study aims to investigate the relationship between maternal serum levels of 25(OH)D in the first trimester and pregnancy and neonatal outcomes. Literature shows that vitamin D promotes extra-villous trophoblast invasion and prevents ischemia-induced endothelial dysfunction, which may lead to the prevention of preeclampsia [14,15]. Vitamin D is also vital in regulating the immune response of the mother to tolerate the developing fetus while fighting infection [16]. Other researchers suggest that early supplementation of vitamin D (from the first trimester) negates the effects of birth season on bone mineral content of neonates [17]. Wagner et al. [18] believed that early pregnancy vitamin D levels might be more important because vitamin D supplementation at later stages may not reduce adverse maternal and neonatal outcomes. The availability of the active 1,25(OH)D depends on the substrate 25(OH)D level. Therefore, we used the first-trimester 25(OH)D level to reflect maternal vitamin D status in this study.

## 2. Materials and Methods

This was a prospective cohort study carried out in a tertiary medical center in Kuala Lumpur, Malaysia. It was a follow-up of a previous study examining the predictors and prevalence of vitamin D deficiency among pregnant mothers [19]. The sample size calculation was based on the prevalence of pregnant women with vitamin D deficiency, where the Z statistic corresponding to the level of confidence was 1.96, the prevalence of vitamin D deficiency derived from a previous study [20] was 0.9, and the precision value (d) was 0.06, using the formula n = Z^2^P(1−P)/d^2^ [21]. All women in their first trimester of pregnancy who attended the antenatal follow-up in the Obstetrics and Gynecology clinic of the tertiary hospital and fulfilled the criteria were invited to participate. Then, 10 mL of maternal blood was taken after informed consent was obtained. The study was approved by the Universiti Kebangsaan Malaysia Medical Research and Ethics Committee (Approval Code: JEP-2018-623). The study was conducted in accordance with the Declaration of Helsinki.

The inclusion criteria were singleton pregnancies between 12–14 weeks gestation and age of mothers between 19–45 years old. The exclusion criteria were multiple pregnancies and pregnant women prescribed with drugs affecting vitamin D metabolism (anti-epileptics, anti-tuberculosis, statins, glucocorticoids).

The subjects’ demographic data were obtained, such as age, ethnicity, parity, previous miscarriages, working status, and pre-pregnancy body mass index (BMI). Underweight was defined as a BMI <18.5 kg/m^2^, normal was 18.5–24.9 kg/m^2^, overweight was 25–29.9 kg/m^2^ and obese was ≥30 kg/m^2^ [22]. Clinical characteristics and pregnancy outcomes of participants were documented, namely associated medical disorders, antenatal complications, mean gestational age at delivery, mode of delivery, type of labor, mean birth weight, mean Apgar score at the 5th minute, neonatal intensive care unit (NICU) admission, and neonatal complications.

### 2.1. Measurement of Maternal Serum 25(OH)D

The venous blood sample collected from each woman at recruitment was centrifuged at 3500 rpm for 10 min at 4 °C (Eppendorf Centrifuge 5810R, Hamburg, Germany). The sera were isolated and stored in −80 °C freezer until analysis. Total serum 25(OH)D levels were measured using enzyme-linked immunosorbent assay (ELISA) (IBL International, Hamburg, Germany) in nmol/L. Depending on the serum 25(OH)D levels, the women were classified to the respective vitamin D status according to the recommendation of the Institute of Medicine (IOM) [7].

### 2.2. Statistical Analysis

Statistical analysis was performed using Statistical Package for the Social Sciences version 25 (IBM, Armonk, NY, USA). Normality of continuous data was assessed using Shapiro–Wilk test. Mean and standard deviation were used to describe continuous variables. Frequency and percentage were used to describe categorical data. Chi-squared test and Fisher’s exact test were used to analyze categorical data for their association. Independent t-test was used to compare 25(OH)D levels between women of different characteristics, pregnancy, and neonatal outcomes. A *p*-value <0.05 was considered statistically significant.

## 3. Results

A total of 60 women, comprising 87%, 7%, 5%, and 2% of Malay, Chinese, Indian, and aborigines were recruited. Their mean age and BMI were 34.8 ± 3.9 years and 26.6 ± 4.9 kg/m^2^. The majority (78%) were multiparous working women (85%) and 26 (43%) of them had at least one previous miscarriage. The mean serum level of 25(OH)D was 34.5 ± 14.1 nmol/L. Overall, 6 (10%) subjects had vitamin D deficiency, 34 (57%) had insufficiency, and 20 (33%) had sufficient 25(OH)D levels based on IOM classification (Table 1).

### 3.1. Serum 25(OH)D Level and Vitamin D Status According to Subjects’ Characteristics

Serum 25(OH)D levels were significantly higher among the Chinese women compared to the Malay women (*p* = 0.049). Otherwise, it did not differ in regard to parity (*p* = 0.55), previous miscarriage (*p* = 0.52), BMI (*p* = 0.24), and working status (*p* = 0.22). Chi-squared analysis did not reveal a significant relationship between vitamin D status (sufficient/insufficient) and subjects’ characteristics (*p* > 0.05) (Table 2). However, BMI (analyzed as continuous data) correlated positively with 25(OH)D levels in pregnant women (r = 0.30, *p* = 0.02).

### 3.2. Serum 25(OH)D Level and Vitamin D Status According to Pregnancy and Neonatal Outcome.

Of the 19 subjects with pregnancy complications, 12 had gestational diabetes mellitus, three has preeclampsia, one had both conditions, and three had other types of complications. 25(OH)D level and vitamin D status were similar between pregnant women with and without antenatal complications and vaginal infection (*p* > 0.05). It was also similar between subjects with different gestation periods (term/preterm/miscarriage), mode of delivery (vaginal/Caesarean), and type of labor (spontaneous/induced) (*p* > 0.05). Of the five neonates with complications, three suffered from respiratory distress syndrome and two from other complications. Serum 25(OH)D level and vitamin D status of the mothers were also similar between neonates with or without complications, neonatal intensive care unit (NICU) admission, or small for gestational age (*p* > 0.05) (Table 3).

## 4. Discussion

This study found a substantial number of pregnant women with vitamin D deficiency and insufficiency as hypothesized. However, no significant difference in serum 25(OH)D level and vitamin D status between women with or without maternal and neonatal complications was found, which defies our expectation.

The high prevalence of vitamin D deficiency and insufficiency among pregnant women in the first trimester was consistent with a previous study reporting that 93% of pregnant women in the third trimester at a hospital in Kuala Lumpur did not have sufficient 25(OH)D levels [10]. Shibata et al. [23] also reported that 90% of the Japanese pregnant women in their study had a 25(OH)D level <20 ng/mL. Another study in Saudi Arabia also reported similar findings, but the population had less sun exposure compared to Malaysian settings due to their regional clothing [24]. The current study showed that Malay subjects had a significantly lower serum 25(OH)D level compared to the Chinese, which was in line with previous findings [25]. This observation might be related to the darker skin pigmentation of the Malays and their fully covered clothing, which reduces sun exposure and synthesis of vitamin D [26].

The current study also did not find a significant relationship between antenatal complications with serum 25(OH)D level/status. The antenatal complications investigated included preeclampsia, gestational diabetes mellitus, and others. Similar to our study, Bodnar et al. [27] did not find a significant association between vitamin D deficiency and hypertensive disorders in pregnancy, including gestational hypertension and preeclampsia. In contrast, several previous studies demonstrated significant associations between low maternal vitamin D status and a higher risk for preeclampsia [11,27,28]. This association is confirmed by a meta-analysis [29]. The discrepancy between the current study and previous studies could arise due to a narrow range of 25(OH)D levels among the subjects of this study. It could also indicate that 25(OH)D levels during the first trimester are not as important as the second and third trimester in dictating the antenatal outcomes of mothers. Several previous studies indicated that vitamin D deficiency was less severe at the later trimesters compared to the first trimester [30,31]. Since we did not measure 25(OH)D levels of our subjects in the second and third trimester, we are not sure whether the same also happened to our subjects. If this happened, it would alter the association between first-trimester 25(OH)D levels and maternal/neonatal outcomes.

Gestational diabetes mellitus (GDM) is a common antenatal complication related to other obstetric problems like Caesarean section rate and preeclampsia [32,33]. Our study did not observe a significant relationship between serum 25(OH)D level with the occurrence of GDM. This finding was in line with an investigation involving a multi-ethnic population in Norway, whereby no association was found between serum maternal 25(OH)D level and risk for gestational diabetes [29]. On the other hand, maternal serum 25(OH)D_3_ level in early pregnancy was inversely associated with GDM risk in a study in Washington, USA [34]. A 5 ng/mL increase in 25(OH)D_3_ concentration was associated with a 14% decrease in GDM risk [34]. Likewise, a meta-analysis of observational studies comprising 22,000 women concluded that maternal vitamin D deficiency was associated with increased GDM risk and infants with low birth weight [35].

The current study also reported no significant relationship between serum maternal 25(OH)D levels and gestational period and mode of delivery. This observation agrees with a recent systematic review by De-Regil et al. [36] which showed that vitamin D supplementation during pregnancy did not reduce the risk of Caesarean section. In contrast, Caesarean delivery was reported to be higher in women with vitamin D deficiency according to a study by Bukhary et al. [20]. Bodnar et al. [37] reported that the risk of preterm delivery (<37 weeks of gestation) was significantly decreased with maternal serum levels of 25(OH)D at approximately 36 ng/mL.

We also did not observe any relationship between neonatal outcomes and the level of maternal 25(OH)D levels. This observation was consistent with a study by Zhou et al. [38], which reported no statistically significant difference in 25(OH)D levels in relation with birth weight, Apgar score at the 5th minute, neonatal jaundice, and NICU admission. Similarly, a meta-analysis showed that vitamin D supplementation during pregnancy did not improve birth weight and birth length of the neonates [39]. In contrast, maternal serum 25(OH)D levels ≥40 ng/mL predicted a 62% lower rate of preterm births compared to those with a level <20 ng/mL at an urban hospital in the US.

Although vitamin D deficiency is prevalent among pregnant women, the relationship of vitamin D status and pregnancy outcomes remains unclear and previous reports are inconsistent. These discrepancies could arise due to the methods of measuring vitamin D and the myriad of confounding variables among the observational studies, including ethnicities and baseline 25(OH)D levels of the subjects [40]. Other researchers found that vitamin D receptor also plays an important role in determining maternal and neonatal health [41,42]. Vitamin D receptor polymorphisms related to FokI, ApaI, and TaqI were reported to be associated with gestational diabetes mellitus, pre-preeclampsia, and preterm birth (reviewed in [43]).

To date, clinical guidelines for vitamin D supplementation cannot be established based on high-quality evidence. It is also not clear if the required intake for pregnant women differs from non-pregnant women. Long-term safety data of vitamin D supplementation in pregnant women has not been established and overdosing of vitamin D might have unfavorable effects, especially in mothers and newborns with mutations involved in vitamin D metabolism [40]. Other researchers hold the view that such worries are unwarranted, and supplementation up to 4000 International Unit vitamin D per day is required to achieve 100 nmol/mL (40 ng/mL) serum 25(OH)D for pregnant mothers to avoid complications [44,45].

Our study has several limitations to be addressed. It is a single center study in an urbanized area of Malaysia, and the non-Malay women might be under-represented. Thus, the readers should generalize the findings of the study with caution. The sample size of this study is small, as the number of participants was based on an earlier study of vitamin D deficiency among pregnant women [19]. Thus, the study might be underpowered for some outcomes. We did not assess whether the subjects had Crohn’s disease or celiac disease, which might impair their vitamin D status. We also did not test the 25(OH)D levels at multiple time points throughout the study period, so we are not sure that 25(OH)D level in the first trimester was reflective of the vitamin D status near the end of the pregnancy period. The subjects might have changed their dietary and sunlight exposure after the initial assessment, and the study may not have captured these changes. From the literature, the changes of 25(OH)D across pregnancy period vary according to studies, whereby increases [30,31], decreases [46], and little change [47] have been reported. The latest study from a Danish Caucasian cohort reported a non-linear trend for 25(OH)D level during pregnancy, whereby it peaked during gestational week 21–34 and declined gradually [48]. In contrast, previous studies invariably reported an increase in 1,25(OH)D level during pregnancy [46,49]. This is attributed to its production from fetal and placental tissues [50]. Increased 1-α-hydroxylase expression in the kidney during pregnancy also contributed to the increase in 1,25(OH)D level [51], but it was not measured in our study.

In contrast, 1,25(OH)D level increased during pregnancy [46,49], probably owing to its production in fetal tissue and placenta [50]. Increased 1α-hydroxylase expression in the kidneys during pregnancy, which converts 25(OH)D to 1,25(OH)D, also contributes to the elevation of 1,25(OH)D level [51]. Although the second and third trimester 25(OH)D levels are closer to pregnancy/delivery events, the influence of vitamin D starts since placentation [52]. Other researchers argued that early pregnancy 25(OH)D levels might be more important because vitamin D supplementation at later stages might not reduce adverse maternal and neonatal outcomes [18]. The seasonal variation of 25(OH)D levels was not considered in this study because Malaysia is a tropical country with relatively consistent sunshine throughout the year (6–7 h sunshine daily) [53]. Therefore, the time of sampling due to variation of sunlight might not constitute a major confounding factor in this study. Nevertheless, the study fills the void of information regarding the relationship between vitamin D status and pregnancy and neonatal outcome in Malaysia.

## 5. Conclusions

Vitamin D deficiency and insufficiency are prevalent among pregnant women in their first trimester. However, there is no significant relationship between 25(OH)D level, vitamin D status, and adverse maternal and neonatal outcomes. This observation needs to be validated in a more extensive and comprehensive prospective study in Malaysia.

## Figures and Tables

**Table 1 ijerph-17-03291-t001:** Demographic data and vitamin D status of the study population (*n* = 60).

Characteristics	*n* (%)
**Ethnicity**	
Malay	52 (87)
Chinese	4 (7)
Indian	3 (5)
Aborigines	1 (2)
**Parity**	
Nulliparous	13 (22)
Multiparous	47 (78)
**Previous miscarriage**	
No	34 (57)
Yes	26 (43)
**Working status**	
Yes	51 (85)
No	9 (15)
**Pre-pregnancy body mass index status**	
Underweight (<18 kg/m^2^)	1 (2)
Normal (18–24.9 kg/m^2^)	25 (42)
Overweight (25–29.9 kg/m^2^)	17 (28)
Obese (>30 kg/m^2^)	17 (28)
**Vitamin D status**	
Deficiency (<30 nmol/L)	6 (10)
Insufficiency (30–50 nmol/L)	34 (57)
Sufficiency (>50 nmol/L)	20 (33)
**Characteristics**	**Mean ± SD (range)**
Age (years)	34 ± 4 (25–42)
Pre-pregnancy body mass index (kg/m^2^)	26.6 ± 4.9 (17.5–37.3)
Serum 25-hydroxyvitamin D (nmol/L)	34.5 ± 14.1 (13.9–76.7)

**Table 2 ijerph-17-03291-t002:** Serum 25-hydroxyvitamin D levels and vitamin D status according to subjects’ characteristics.

Characteristic	25-hydroxyvitamin D (nmol/L)	*p*	Vitamin D Status		*p*
	Mean ± SD		Sufficient	Insufficient	
**Ethnicity**		**0.049**			0.55
Malay and aborigines (*n* = 53)	35.4 ± 13.7 *		5	48	
Chinese (*n* = 4)	53.0 ± 15.9 *		1	3	
Indian (*n* = 3)	33.2 ± 5.7		0	3	
**Parity**		0.55			0.75
nulliparous (*n* = 13)	34.4 ± 9.8		1	12	
multiparous (*n* = 47)	37.1 ± 15.2		5	42	
**Previous miscarriage**		0.52			0.73
No (*n* = 34)	35.4 ± 13.7		3	31	
Yes (*n* = 26)	37.8 ± 14.8		3	23	
**Pre-pregnancy** **body mass index**		0.24			0.11
<25 kg/m^2^ (*n* = 26)	34.5 ± 12.8		1	25	
25–29.9 kg/m^2^ (*n* = 17)	34.6 ± 14.3		1	16	
>30 kg/m^2^ (*n* = 17)	41.4 ± 15.5		4	13	
**Working status**		0.22			1.00
No (*n* = 9)	41.8 ± 12.0		1	8	
Yes (*n* = 51)	35.5 ± 14.4		5	46	
**Existing comorbidities**		0.24			0.35
No (*n* = 42)	35.1 ± 13.3		3	39	
Yes (*n* = 18)	39.7 ± 15.9		3	15	

*—indicates a significant difference between the marked groups. The *p*-value < 0.05 was bolded.

**Table 3 ijerph-17-03291-t003:** Serum 25-hydroxyvitamin D levels and vitamin D status based on maternal and neonatal outcomes.

Characteristic	25-hydroxyvitamin D (nmol/L)	*p*	Vitamin D Status	*p*
Mean ± SD	Sufficient	Insufficient
**Antenatal complications**		0.37			0.98
No (*n* = 41)	36.7 ± 14.8		5	27	
Gestational diabetes mellitus (*n* = 13)	39.1 ± 13.4		1	9	
Others (*n* = 6)	29.3 ± 10.0		0	4	
**Vaginal infection**		0.96			0.68
No (*n* = 50)	36.8 ± 14.1		4	45	
Yes (*n* = 5)	37.2 ± 18.2		1	4	
**Gestation at birth**		0.23			0.31
Term (≥37 weeks) (*n* = 43)	34.5 ± 14.5		3	40	
Preterm (24–36^+6^ weeks) (*n* = 13)	42.1 ± 13.5		2	11	
Miscarriage (<24 weeks) (*n* = 4)	39.3 ± 8.8		1	3	
**Mode of delivery**		0.43			1.00
Vaginal or instrumental (*n* = 34)	35.0 ± 15.2		3	31	
Caesarean (*n* = 22)	38.2 ± 13.4		2	20	
**Type of labor**		0.11			0.406
Spontaneous (*n* = 51)	37.3 ± 14.7		5	46	
Induced (*n* = 5)	26.4 ± 6.2		1	3	
**Neonatal intensive care unit (** **NICU) admission**		0.09			0.50
No (*n* = 49)	35.0 ± 14.7		4	45	
Yes (*n* = 7)	44.9 ± 9.5		1	6	
**Neonatal complications**		0.28			0.38
No (*n* = 51)	35.6 ± 14.6		4	47	
Yes (*n* = 5)	43.0 ± 12.7		1	4	
**Apgar score at the 5th minutes**		0.31			0.17
<7 (*n* = 2)	46.7 ± 23.6		4	50	
≥7 (*n* = 54)	35.9 ± 14.3		1	1	
**Small for gestational age**		0.80			1.00
No (*n* = 53)	36.4 ± 14.7		5	48	
Yes (*n* = 3)	34.1 ± 11.5		0	3

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
