# Peer review of "Is First Trimester Maternal 25-Hydroxyvitamin D Level Related to Adverse Maternal and Neonatal Pregnancy Outcomes? A Prospective Cohort Study among Malaysian Women"

_ijerph, 2020, doi:10.3390/ijerph17093291_

Round 1
Reviewer 1 Report
The authors performed a prospective cohort study on a sample of 60 pregnant women. The study investigated the associations between maternal serum 25-hydroxyvitamin D levels in the first trimester of pregnant women and their pregnancy/neonatal outcomes. They observed 25(OH)D was not associated with adverse maternal and neonatal outcomes.
The major limitations of this study is a poor sample size and a single check for 25(OH)D.
I suggest removing to the Discussion Section the last two period of the Introduction.
Season can affect vitamin D levels. Did you consider this particular feature (alternatively comment this point in the discussion).
Did the authors detect any correlation between BMI (as for continuous variable) and 25(OH)D?
May vitamin D status change over pregnancy time due to sun exposure or diet modification during pregnancy? Did the recruited pregnant women take any vitamin D supplement or multivitamins?
A large amount of women reported gestational diabetes. Do you consider separately this group of women from the other women as for 25(OH)D levels and related outcomes?
Is it possible the authors speculate about the role of VDR?
Minor points:
Line 149: maternal
Author Response
Thank you for reviewing our manuscript. We value your constructive comments and they have been addressed point-to-point in the attached response sheet.
We thank you in advance for re-assessing our manuscript.
Title: Is first trimester maternal 25-hydroxyvitamin D level related to adverse maternal and neonatal pregnancy outcomes? A prospective cohort study among Malaysian women
ID: ijerph-778758
Thank you for reviewing our manuscript. We appreciate your valuable comments and they are responded as the following:
Reviewer 1
|
Comment |
Reply |
|
The major limitations of this study is a poor sample size and a single check for 25(OH)D.
|
Thank you for your comment. The sample size was derived from our prior study on vitamin D deficiency prevalence in pregnant women, and this information has been added in Methods (Line 65-68). We acknowledge that the sample size is limited and one-time testing of the vitamin D in the discussion as limitations (Line 198-200). |
|
I suggest removing to the Discussion Section the last two period of the Introduction. |
Thank you for your suggestion. We have moved the last two sentences of Introduction to the beginning of the Discussion (Line 132-135). |
|
Season can affect vitamin D levels. Did you consider this particular feature (alternatively comment this point in the discussion). |
Thank you for your suggestion. The seasonal variation of vitamin D level was not considered in this study because Malaysia is a tropical country with relatively consistent sunshine throughout the years (6-7 hours sunshine daily). Therefore, the time of sampling due to variation of sunlight might not constitute a major confounding factor in this study. This point was added to the discussion section (Line 205-207). |
|
Did the authors detect any correlation between BMI (as for continuous variable) and 25(OH)D? |
Thank you for your suggestion. We detected a positive correlation between BMI and vitamin D level when BMI was analysed as a continuous variable. This point is added to the results (Line 114). |
|
May vitamin D status change over pregnancy time due to sun exposure or diet modification during pregnancy? Did the recruited pregnant women take any vitamin D supplement or multivitamins? |
Thank you for you comment. We acknowledge that the vitamin D level might change due to modification in dietary and time of sunshine exposure. The lack of assessment with these aspects has been acknowledged in the limitations (Line 202-204). |
|
A large amount of women reported gestational diabetes. Do you consider separately this group of women from the other women as for 25(OH)D levels and related outcomes? |
Thank you for your suggestion. We have separate gestational diabetes from other antenatal complications in Table 3 and analyse them separately. |
|
Is it possible the authors speculate about the role of VDR? |
Thank you for your suggestion. We added that ‘…Other researchers found that vitamin D receptor also plays an important role in determining maternal and neonatal health [33,34]. Vitamin D receptor polymorphisms related to FokI, ApaI and TaqI were reported to be associated with gestational diabetes mellitus, preeclampsia and preterm birth (reviewed in [35]). (Line 184-187)’ |
|
Line 149: maternal |
Thank you for your reminder. We have placed the word ‘material’ to ‘maternal’ |
Thank you for your time in reviewing our manuscript again.
Reviewer 2 Report
The article entitled "Is first trimester maternal 25-hydroxyvitamin D level related to adverse maternal and neonatal pregnancy outcomes? A prospective cohort study among Malaysian women" (ijerph-778758).
The onjetive of this article is to determine the relationship between the level of maternal 25-hydroxyvitamin D in the first trimester of pregnant women and their pregnancy / neonatal outcomes.
Comments:
Summary: More clarity, for example vitamin D levels are determined in the first trimester.
Introduction: It is scarce and should be broadened and focused more specifically on the pregnancy object of this work.
Material and methods: Presents approval of the ethical committee.
It is not specified whether the pregnant women took polyvitamins during pregnancy. This is quite common and can be a major bias in the study. This aspect should be clarified.
Sample size is not justified.
Mothers' different ethnicity can influence diet and in turn vitamin D levels.
Results:
Table 1.- Add comparison of proportions (p-value).
Discussion: Study limitations such as unwarranted sample size are not assessed. Very small in pregnancy studies. Pathologies that may occur during pregnancy and that may interfere with the outcome of the pregnancy are not assessed.
Author Response
Thank you for reviewing our manuscript. We value your constructive comments and they have been addressed point-to-point in the attached response sheet.
We thank you in advance for re-assessing our manuscript.
Title: Is first trimester maternal 25-hydroxyvitamin D level related to adverse maternal and neonatal pregnancy outcomes? A prospective cohort study among Malaysian women
ID: ijerph-778758
Thank you for reviewing our manuscript. We appreciate your valuable comments and they are responded as the following:
Reviewer 2
|
Comment |
Reply |
|
Summary: More clarity, for example vitamin D levels are determined in the first trimester. |
Thank you for your comment. We have added in the abstract that “Their blood was collected in the first trimester for total serum 25-hydroxyvitamin D determination”. (Line 21-22) |
|
Introduction: It is scarce and should be broadened and focused more specifically on the pregnancy object of this work. |
Thank you for your comment. Paragraph 2 of the introduction focuses on the prevalence of vitamin D deficiency in pregnant women and its effects on their health and neonatal outcomes. We have further enhanced the paragraph by indicating the molecular action of vitamin D on women and children health as the following: ‘…Vitamin D affects material and neonatal health by influencing gene expression related to immunity and inflammation in mothers, and DNA methylation and epigenetic regulation in children. Some researchers believed that women need to enter pregnancy with sufficient vitamin D level to prevent complications like pre-eclampsia, and later stage supplementation is futile in preventing pregnancy complications (Hollis and Wagner 2017).’ (Line 54-56) |
|
Material and methods: Present approval of the ethical committee. |
Thank you for your comments. We have stated that the present project was approved by Universiti Kebangsaan Malaysia Medical Research and Ethics Committee (Approval Code: JEP-2018-623) (Line 71-72) |
|
It is not specified whether the pregnant women took polyvitamins during pregnancy. This is quite common and can be a major bias in the study. This aspect should be clarified. |
Thank you for your comments. We did not trace the changes in vitamin D intake and behaviour changes throughout the entire course of the study. This limitation has been acknowledged in the discussion section. (Line 201-205) |
|
Sample size is not justified. |
Thank you for your comment. The sample size was derived from our prior study on vitamin D deficiency prevalence in pregnant women, and this information has been added in the Methods (Line 65-68). We acknowledge that the sample size is limited and one-time testing of the vitamin D in the discussion as a limitation. (Line 198-200) |
|
Mothers' different ethnicity can influence diet and in turn vitamin D levels. |
Thank you for your comment. We ask for your kind consideration for this comment. If the mother’s ethnicity influenced the serum vitamin D level, the difference would have been detected among subjects of different ethnicities. |
|
Results: Table 1.- Add comparison of proportions (p-value). |
Thank you for your comment. We ask for your kind reconsideration for this comment because comparison of proportions requires at least a 2×2 design. If the reviewer is asking for a comparison of proportion with regards to vitamin D status, the data have been presented in Table 2. |
|
Discussion: Study limitations such as unwarranted sample size are not assessed. Very small in pregnancy studies. Pathologies that may occur during pregnancy and that may interfere with the outcome of the pregnancy are not assessed. |
Thank you for your comment. We acknowledge that we did not conduct clinical examinations on the patients for Chron disease or celiac disease during recruitment. These conditions might impair their vitamin D status. The limited sample size was also addressed in the limitation paragraph. (Line 200-201) |
Thank you for your time in reviewing our manuscript again.
Reviewer 3 Report
Significant digits. The general rule is that no more non-zero digits should be given than are justified by the uncertainty of the value.
See "Too many digits: the presentation of numerical data"
https://www.ncbi.nlm.nih.gov/pmc/articles/PMC4483789/
If the uncertainty is greater than about 7%, only two non-zero digits are justified.
P values should be given to two decimal places unless the first two are 00 or the number lies between 0.045 and 0.050.
Thus,
|
Malay and aborigines |
35.40 ± 13.72* |
Should be
|
Malay and aborigines |
35 ± 14* |
|
Age (years) |
34.4 ± 3.9 (25 – 42) |
Should be
|
Age (years) |
34 ± 4 (25 – 42) |
|
Malay |
52 (86.7) |
With 60 participants, % should be in whole numbers
Please review all numbers in abstract, text, tables, and figures and adjust accordingly.
Comment: Should be 25-hydroxyvitamin D [25(OH)D] at first mention, 25(OH)D thereafter if used at least three times in abstract, same for text. Vitamin D status is OK.
138 The current study also did not find a significant relationship between antenatal complications
139 with serum 25-hydroxyvitamin D level/status.
Comment: The likely reasons include that the range of 25OHD was too low and that first trimester 25OHD may not be as important as second and third trimester values.
168 Despite that vitamin D deficiency is prevalent among pregnant women, the relationship of
169 vitamin D status and pregnancy outcomes remains unclear and previous reports were inconsistent
This article should be discussed
Maternal 25(OH)D concentrations ≥40 ng/mL associated with 60% lower preterm birth risk among general obstetrical patients at an urban medical center.
McDonnell SL, Baggerly KA, Baggerly CA, Aliano JL, French CB, Baggerly LL, Ebeling MD, Rittenberg CS, Goodier CG, Mateus Niño JF, Wineland RJ, Newman RB, Hollis BW, Wagner CL.
PLoS One. 2017 Jul 24;12(7):e0180483.
And one or more of the articles and reviews by Hollis and Wagner should be discussed
The Implications of Vitamin D Status During Pregnancy on Mother and her Developing Child.
Wagner CL, Hollis BW.
Front Endocrinol (Lausanne). 2018 Aug 31;9:500.
New insights into the vitamin D requirements during pregnancy.
Hollis BW, Wagner CL.
Bone Res. 2017 Aug 29;5:17030.
Vitamin D supplementation during pregnancy: Improvements in birth outcomes and complications through direct genomic alteration.
Hollis BW, Wagner CL.
Mol Cell Endocrinol. 2017 Sep 15;453:113-130.
Vitamin D supplementation during pregnancy: double-blind, randomized clinical trial of safety and effectiveness.
Hollis BW, Johnson D, Hulsey TC, Ebeling M, Wagner CL.
J Bone Miner Res. 2011 Oct;26(10):2341-57.
Author Response
Thank you for reviewing our manuscript. We value your constructive comments and they have been addressed point-to-point in the attached response sheet.
We thank you in advance for re-assessing our manuscript.
Title: Is first trimester maternal 25-hydroxyvitamin D level related to adverse maternal and neonatal pregnancy outcomes? A prospective cohort study among Malaysian women
ID: ijerph-778758
Thank you for reviewing our manuscript. We appreciate your valuable comments and they are responded as the following:
Reviewer 3
|
Comment |
Reply |
||||||||||
|
Significant digits. The general rule is that no more non-zero digits should be given than are justified by the uncertainty of the value. See "Too many digits: the presentation of numerical data" https://www.ncbi.nlm.nih.gov/pmc/articles/PMC4483789/ If the uncertainty is greater than about 7%, only two non-zero digits are justified. P values should be given to two decimal places unless the first two are 00 or the number lies between 0.045 and 0.050. Thus,
Should be
Should be
With 60 participants, % should be in whole numbers Please review all numbers in abstract, text, tables, and figures and adjust accordingly. |
Thank you for your comment. We agreed to: 1. change all percentage to integer 2. change age (mean and SD) to integer 3. change all other variables (mean and SD) to 1 decimal places. 4. change all p-value to 2 decimal places
|
||||||||||
|
Comment: Should be 25-hydroxyvitamin D [25(OH)D] at first mention, 25(OH)D thereafter if used at least three times in abstract, same for text. Vitamin D status is OK.
|
Thank you for your comment. We have used the abbreviation 25(OH)D to replace 25-hydroxyvitamin D in the abstract and main text. However, we retain 25-hydroxyvitamin D in the Tables because they are read independently from the text. |
||||||||||
|
138 The current study also did not find a significant relationship between antenatal complications Comment: The likely reasons include that the range of 25OHD was too low and that first trimester 25OHD may not be as important as second and third trimester values. |
Thank you for your input. We have added the explanation in the discussion. ‘The discrepancy between the current study and previous studies could arise due to a narrow range of vitamin D levels among the subjects of this study. It could also indicate that vitamin D level during the first trimester is not as important as the second and third trimester in dictating the antenatal outcomes of mothers.’ (Line 152-155) |
||||||||||
|
168 Despite that vitamin D deficiency is prevalent among pregnant women, the relationship of This article should be discussed Maternal 25(OH)D concentrations ≥40 ng/mL associated with 60% lower preterm birth risk among general obstetrical patients at an urban medical center. McDonnell SL, Baggerly KA, Baggerly CA, Aliano JL, French CB, Baggerly LL, Ebeling MD, Rittenberg CS, Goodier CG, Mateus Niño JF, Wineland RJ, Newman RB, Hollis BW, Wagner CL. PLoS One. 2017 Jul 24;12(7):e0180483.
|
Thank you for your suggestion. To elaborate on this point, we added that “…These discrepancies could arise due to the methods of measuring vitamin D, the myriad confounding variables among the observational studies, including the ethnicities and baseline vitamin D level of the subjects.” (Line 182-184) We have added the study by McDonnell et al. as in the paragraph discussing the neonatal outcomes as the following: “In contrast, maternal serum 25(OH)D level ≥ 40 ng/mL predicted 62% lower rate of preterm births compared to those with a level < 20 ng/mL at an urban hospital in the US” (Line 178-179) |
||||||||||
|
And one or more of the articles and reviews by Hollis and Wagner should be discussed The Implications of Vitamin D Status During Pregnancy on Mother and her Developing Child. Wagner CL, Hollis BW. Front Endocrinol (Lausanne). 2018 Aug 31;9:500.
New insights into the vitamin D requirements during pregnancy. Hollis BW, Wagner CL. Bone Res. 2017 Aug 29;5:17030.
Vitamin D supplementation during pregnancy: Improvements in birth outcomes and complications through direct genomic alteration. Hollis BW, Wagner CL. Mol Cell Endocrinol. 2017 Sep 15;453:113-130.
Vitamin D supplementation during pregnancy: double-blind, randomized clinical trial of safety and effectiveness. Hollis BW, Johnson D, Hulsey TC, Ebeling M, Wagner CL. J Bone Miner Res. 2011 Oct;26(10):2341-57.
|
Thank you for your suggestion. We have incorporated the informative references in our articles: Introduction: Vitamin D affects material and neonatal health by influencing gene expression related to immunity and inflammation in mothers, and DNA methylation and epigenetic regulation in children. Some researchers believed that women need to enter pregnancy with sufficient vitamin D level to prevent complications like pre-eclampsia, and later stage supplementation is futile in preventing pregnancy complications (Hollis and Wagner 2017). (Line 56-59) Discussion: … Other researchers hold the view that such worries are unwarranted, and supplementation up to 4000 IU vitamin D per day is required to achieve 100 nmol/mL (40 ng/mL) serum 25(OH)D for pregnant mothers to avoid complications (Hollis and Wagner 2017; Hollis et al. 2011). (Line 192-195) |
Thank you for your time in reviewing our manuscript again.
Round 2
Reviewer 2 Report
After reviewing the new version of the article "Is first trimester maternal 25-hydroxyvitamin D level related to adverse maternal and neonatal pregnancy outcomes? A prospective cohort study among Malaysian women" (ijerph-778758) and the authors' responses to comments made.
I have verified that all suggestions have been incorporated into the manuscript. The understanding of the work and the tables has improved, and the information contained in them is more understandable.
The article has been improved with a greater use of the bibliography on the subject.
Author Response
Comment: I have verified that all suggestions have been incorporated into the manuscript. The understanding of the work and the tables has improved, and the information contained in them is more understandable. The article has been improved with greater use of the bibliography on the subject.
Reply: Thank you for re-assessing our manuscript and your previous constructive comments. There is no other comment to be addressed.
Thank you.
Reviewer 3 Report
Vitamin D level still occurs in at least two locations (in red-font text)
Serum 25(OH)D and 1,25(OH)2D increase during the course of pregnancy.
It would be interesting to know how they change as a function of both in the first trimester.
I would like the authors to search the journal literature for articles on changes in serum 25OHD during the course of pregnancy. One aspect: does change depend on first trimester level? Another aspect: during which trimester are pregnancy and birth outcomes is 25OHD level most important. This exercise is worth doing since it may be that the second or third trimester levels are much more important than the first trimester levels. I imagine readers of the article might have such questions, so it is worthwhile to address them now.
Vitamin D status in pregnant Indian women across trimesters and different seasons and its correlation with neonatal serum 25-hydroxyvitamin D levels.
Marwaha RK, Tandon N, Chopra S, Agarwal N, Garg MK, Sharma B, Kanwar RS, Bhadra K, Singh S, Mani K, Puri S.
Br J Nutr. 2011 Nov;106(9):1383-9.
Association of maternal serum 25-hydroxyvitamin D concentrations in second and third trimester with risk of gestational diabetes and other pregnancy outcomes.
Wen J, Hong Q, Zhu L, Xu P, Fu Z, Cui X, You L, Wang X, Wu T, Ding H, Dai Y, Ji C, Guo X.
Int J Obes (Lond). 2017 Apr;41(4):489-496.
High prevalence of vitamin D deficiency in pregnant Korean women: the first trimester and the winter season as risk factors for vitamin D deficiency.
Choi R, Kim S, Yoo H, Cho YY, Kim SW, Chung JH, Oh SY, Lee SY.
Nutrients. 2015 May 11;7(5):3427-48. doi: 10.3390/nu7053427.
Author Response
Dear reviewer,
We are grateful for your review and insightful comments.
Title: Is first trimester maternal 25-hydroxyvitamin D level related to adverse maternal and neonatal pregnancy outcomes? A prospective cohort study among Malaysian women
ID: ijerph-778758
We are grateful for your review and insightful comments. They are addressed as the following:
Reviewer 3
|
Comment |
Reply |
|
Vitamin D level still occurs in at least two locations (in red-font text) |
Thank you for the reminder. We have changed the “vitamin D level” to “25(OH)D level” wherever it is appropriate. |
|
Serum 25(OH)D and 1,25(OH)2D increase during the course of pregnancy. It would be interesting to know how they change as a function of both in the first trimester. I would like the authors to search the journal literature for articles on changes in serum 25OHD during the course of pregnancy. One aspect: does change depend on first trimester level? Another aspect: during which trimester are pregnancy and birth outcomes is 25OHD level most important. This exercise is worth doing since it may be that the second or third trimester levels are much more important than the first trimester levels. I imagine readers of the article might have such questions, so it is worthwhile to address them now. Vitamin D status in pregnant Indian women across trimesters and different seasons and its correlation with neonatal serum 25-hydroxyvitamin D levels. Marwaha RK, Tandon N, Chopra S, Agarwal N, Garg MK, Sharma B, Kanwar RS, Bhadra K, Singh S, Mani K, Puri S. Br J Nutr. 2011 Nov;106(9):1383-9. Association of maternal serum 25-hydroxyvitamin D concentrations in second and third trimester with risk of gestational diabetes and other pregnancy outcomes. Wen J, Hong Q, Zhu L, Xu P, Fu Z, Cui X, You L, Wang X, Wu T, Ding H, Dai Y, Ji C, Guo X. Int J Obes (Lond). 2017 Apr;41(4):489-496. High prevalence of vitamin D deficiency in pregnant Korean women: the first trimester and the winter season as risk factors for vitamin D deficiency. Choi R, Kim S, Yoo H, Cho YY, Kim SW, Chung JH, Oh SY, Lee SY. Nutrients. 2015 May 11;7(5):3427-48. doi: 10.3390/nu7053427.
|
Thank you for the suggestion. We have incorporated the references suggested and added the following statement: “From the literature, the circulating 25(OH)D level was reported to increase [39] or decrease marginally [40], or show negligible change during pregnancy [41]. In contrast, 1,25(OH)D level increases during pregnancy [40,42], probably owing to its production in foetal tissue and placenta [43]. Increased 1α-hydroxylase expression in the kidneys during pregnancy, which converts 25(OH)D to 1,25(OH)D, also contributes to the elevation of 1,25(OH)D level [44]. Although the second and third trimester vitamin D levels are closer to pregnancy/delivery events, the influence of vitamin D starts since placentation [45]. Other researchers argued that early pregnancy vitamin D level might be more important because vitamin D supplementation at later stages might not reduce adverse maternal and neonatal outcomes [46].” |
Thank you for reassessing our manuscript.